# Amur Tiger Individual Identification Based on the Improved InceptionResNetV2

**DOI:** 10.3390/ani14162312

**Published:** 2024-08-09

**Authors:** Ling Wu, Yongyi Jinma, Xinyang Wang, Feng Yang, Fu Xu, Xiaohui Cui, Qiao Sun

**Affiliations:** 1School of Information and Technology (School of Artificial Intelligence), Beijing Forestry University, Beijing 100083, China; 202321081058@mail.bnu.edu.cn (L.W.); yy1746228151@gmail.com (Y.J.); fengyang@bjfu.edu.cn (F.Y.); cuixiaohui@bjfu.edu.cn (X.C.); sunqiao@bjfu.edu.cn (Q.S.); 2School of Artificial Intelligence, Beijing Normal University, Beijing 100875, China; 3Engineering Research Center for Forestry-Oriented Intelligent Information Processing of National Forestry and Grassland Administration, Beijing 100083, China; 4State Key Laboratory of Efficient Production of Forest Resources, Beijing 100083, China

**Keywords:** convolutional neural network, object detection, individual recognition, InceptionResNetV2, attention mechanism, 68T07, 68U10, 92B20

## Abstract

**Simple Summary:**

Accurate identification of individual Amur tigers is vital for their conservation, as it helps us understand their population and distribution. Existing identification methods often fall short in accuracy, and our study focuses on creating a more accurate method for identifying individual Amur tigers using advanced deep learning techniques. We improved an existing neural network model called InceptionResNetV2 by adding features like dropout layers and dual-attention mechanisms to better capture the unique stripe patterns of each tiger and reduce errors during training. We tested our model on a large dataset of tiger images and found it to be highly effective, achieving an average recognition accuracy of over 95% for different body parts, with left stripes reaching the highest 99.37%. This method significantly outperforms previous models and provides a reliable tool for wildlife researchers and conservationists to monitor and protect Amur tigers. By improving the ability to track individual tigers, our research offers practical benefits for preserving this endangered species and enhancing wildlife management practices globally.

**Abstract:**

Accurate and intelligent identification of rare and endangered individuals of flagship wildlife species, such as Amur tiger (*Panthera tigris altaica*), is crucial for understanding population structure and distribution, thereby facilitating targeted conservation measures. However, many mathematical modeling methods, including deep learning models, often yield unsatisfactory results. This paper proposes an individual recognition method for Amur tigers based on an improved InceptionResNetV2 model. Initially, the YOLOv5 model is employed to automatically detect and segment facial, left stripe, and right stripe areas from images of 107 individual Amur tigers, achieving a high average classification accuracy of 97.3%. By introducing a dropout layer and a dual-attention mechanism, we enhance the InceptionResNetV2 model to better capture the stripe features of individual tigers at various granularities and reduce overfitting during training. Experimental results demonstrate that our model outperforms other classic models, offering optimal recognition accuracy and ideal loss changes. The average recognition accuracy for different body part features is 95.36%, with left stripes achieving a peak accuracy of 99.37%. These results highlight the model’s excellent recognition capabilities. Our research provides a valuable and practical approach to the individual identification of rare and endangered animals, offering significant potential for improving conservation efforts.

## 1. Introduction

Wildlife plays a crucial role in maintaining ecological balance and stability [1]. However, the continuous expansion of human activities in recent years has led to significant encroachment and destruction of animal habitats. This has pushed many species to the brink of extinction or into a critically endangered state, posing a serious threat to biodiversity both in China and globally [2]. It is imperative to implement effective and feasible conservation measures to address this crisis. Among them, precise monitoring and identification of animal species and individuals is an extremely effective means of understanding the characteristics of animal populations [3].

Recently, studies showed that individual identification is becoming more important, since it helps us to understand the density, space use, behaviors, etc., of different animal species [4] (Roy et al., 2023 [5]; Akçay et al., 2020 [6]; Dave et al., 2023 [7]; Schütz et al., 2021 [8]; Xu et al., 2024 [9]). Animal individual identification is not only a necessary basis for unbiased data collection (such as recording individual behavior data), but also for recording individual changes in the variables of concern (social relations, special behaviors, etc.) [10]. Therefore, the identification of individual wild animals is conducive to a profound understanding of the ecological habits, population distribution, breeding status, and habitat status of animal populations. There are various methods for identifying individual animals, including vocalization [11,12], fur color [13], gait [14,15], and DNA from body tissues [16,17], among which using the appearance of animals such as fur for individual identification is a very common and effective method [18]. For animals with distinct fur stripes, such as Amur tigers [19] and zebras [20,21,22,23], using the differences in individual stripes to identify different animals has become a hot topic for researchers. Taking Amur tigers as an example, they usually have unique black stripes on their bodies, which have high uniqueness and invariance. Individual identification can be made by comparing the stripes in different areas of the body. At present, this method is one of the most important means to distinguish the individual tigers in the northeast region. Additionally, akin to human faces, each Amur tiger has unique facial features, which can also be used for individual identification.

In recent years, due to the strong feature extraction capabilities of deep learning technologies such as convolutional neural networks (CNNs), and their high recognition efficiency and accuracy, they have gradually become widely used for the intelligent identification of animal species and individuals. For example, in species identification, in 2014, Chen et al. [24] used a CNN for the first time to classify 20 species in 20,000 pictures. In 2016, Gomez et al. [25] trained a network on the African Serengeti wildlife dataset SSe. In 2017, Cheema et al. [26] used Faster R-CNN to detect species with different patterns (such as tigers, zebras and other individuals) in pictures. In 2018, Norouzzadeh et al. [27] realized the automatic classification, quantity detection and behavior description of 48 species, with an accuracy rate of 92%. In 2019, Gong Yinan et al. [28] carried out automatic species identification from infrared camera images of Amur tigers and leopards taken under natural conditions based on the YOLOv3 model, and the average accuracy rate of identification of eight species reached 84.9~96.0%. There are also many applications based on CNNs for individual animal identification. For example, in 2018, Fan et al. [29] applied the improved convolutional neural network model GKP-Net to analyze the facial features of 48 golden monkeys, with an accuracy up to 93.69%. In 2018, Zhao et al. [30] built an individual recognition model for three leopards based on the Cifar-10 model, with an accuracy rate up to 99.3%. In 2020, He et al. [3] extracted panda individual facial features with YOLOv3 and a CNN, with an accuracy up to 98%.

Additionally, there is research that specifically focuses on individual Amur tiger recognition with CNNs. For example, in 2020, Shi et al. [19] built a nine-layer deep convolutional neural network framework and identified 40 individual Amur tigers based on the striped features of the left and the right sides of individual animals, with an accuracy rate of 93.5%. In 2021, Shi et al. [31] applied several classical convolutional neural networks to recognize the different body parts of 38 Amur tiger individuals, with an accuracy rate of 97.01%. However, these studies are all based on independently collected Amur tiger stripe data, but the data sample size is very limited due to the rareness of the Amur tiger, and the existing datasets have not been publicly verified, making it difficult to objectively evaluate the effectiveness and performance of the algorithms. Furthermore, constrained by factors such as image resolution, lighting, clarity, obstruction levels, and the posture and behavior of the Amur tigers, these studies did not present Amur tiger photos taken in the wild environment. Meanwhile, the network models and methods used in these studies are relatively primitive. For example, the literature [19] mainly employed a simple layer-stacking approach, while the literature [30] directly used classic convolutional neural networks, such as LeNet, AlexNet, ZFNet, VGG16, and ResNet34, rather than more advanced networks. These networks need further improvement in aspects such as feature extraction capability, overfitting suppression, and multi-scale image recognition.

In view of the above reasons, this paper conducts individual recognition based on the different physiological characteristics of Amur tigers and builds an individual recognition network based on the improved InceptionResNetV2 model. The InceptionResNet model combines residual blocks with an Inception module to decompose large-scale convolution kernels into multiple small-scale convolution kernels for dimensionality reduction, which can achieve a balance between network width and depth while retaining rich feature expression capability. At the same time, the residual connection is used to deepen the network and accelerate the convergence speed. However, if the InceptionResNet model is directly applied to the Amur tiger individual recognition problem, it may lead to insufficient expression ability and overfitting. Therefore, this paper introduces a dropout layer and a dual-attention mechanism module to improve model recognition accuracy and loads pre-trained models with transfer learning to further improve model training efficiency. It provides an important reference value for the accurate identification of wildlife images and further improves the intelligence level of wildlife monitoring.

The rest of this paper is organized as follows. Section 2 detects and segments the facial, left stripe, and right stripe areas from Amur tiger images with YOLOv5. Section 3 introduces the improved InceptionResNetV2 model. Section 4 verifies the validity of our proposed method by experiments. Section 5 summarizes this paper.

## 2. Amur Tiger Stripe Part Detection with YOLOV5

### 2.1. Material and Data Preparation

In this paper, we utilize the large wild animal ATRW dataset originally published as part of the re-identification challenge at the ICCV 2019 Workshop on Computer Vision for Wildlife Conservation (available from https://lila.science/datasets/atrw, accessed on 24 February 2024), which serves as the benchmark for Amur tiger re-identification in the wild. It contains a total of 3393 images of 107 Amur tigers from various wildlife parks (there are fewer than 600 wild tigers in the world), of which 1887 images have been tagged with IDs. The researchers used time-synchronized surveillance cameras and tripod-mounted SLRs to capture images in an unconstrained environment. A total of 8076 high-resolution (1920 × 1080) video clips were captured [32], which were further evenly sampled into frames, some of which were discarded due to ghosting artifacts, lack of tigers, or the presence of other noise. In general, the dataset is large and well annotated, which is sufficient for model training and testing.

To facilitate object detection tasks, we utilized LabelImg to meticulously annotate the distinct features of each Amur tiger’s photo in the dataset. Specifically, we labeled the facial region, as well as the stripes that appear on the left side of the tiger when viewed from the front (referred to as “left stripes”) and those on the right side (referred to as “right stripes”). After annotation, we converted the dataset into COCO format, which is supported by the YOLOv5 model. Since most images contain the faces of Amur tigers, but there are very few instances where both left and right stripes appear simultaneously, it is challenging to maintain a consistent number of annotations for the face and the other two parts. Thus, we endeavored to balance the number of annotations for the left and right stripes. A total of 241 images were annotated in this research, with 210 annotations for faces, 106 for left stripes, and 108 for right stripes. Ultimately, the annotated images were partitioned into training, validation, and test sets in an 8:1:1 ratio.

### 2.2. Stripe Part Detection and Result Analysis

To achieve accurate identification of the faces and stripes on both sides of Siberian tigers, this study employs the YOLOv5 model [33] to perform target detection on different parts of the Amur tiger in the dataset, and obtains images of faces, left stripes, and right stripes through operations such as bounding and cropping. The settings for the model training parameters are as follows:(1)Number of iterations: epoch = 300;(2)Batch size: batch_size = 16;(3)Initial learning rate: lr0 = 0.01 (dynamically reduce the learning rate using a cosine function)(4)Optimizer: optimizer = SGD

The effectiveness of the object detection in this study is assessed from several aspects including training loss, validation index, object detection, and segmentation results.

#### 2.2.1. Training Loss

We choose the rectangle box loss, objectness loss, and classification loss to objectively evaluate the performance and stability of the target detection model during the training process. The loss function curves are shown in Figure 1. As can be seen from the figure, the three losses basically maintain a trend of gradual decline, and the final loss value is below 0.02.

#### 2.2.2. Validation Index

To evaluate the recognition performance, we generated a confusion matrix using the trained weight file, as shown in Figure 2a. As illustrated in the figure, the diagonal accuracy rates are 95%, 100%, and 80%, respectively, suggesting effective object detection performance.

Based on the changes in accuracy and recall during the training process, the P-R curve is depicted in Figure 2b. By calculating the area enclosed by the P-R curve and the coordinate axes, the mAP@0.5 reaches 97.3%, indicating excellent training results. This validates the effectiveness and accuracy of the Amur tiger physiological feature detection model applied in this paper.

#### 2.2.3. Detection and Segmentation Results

The well-trained YOLOv5 model, when applied to the entire dataset, can effectively detect various physiological features of the Amur tiger and perform target framing and category labeling. Several images were randomly selected from the validation set and annotated by the target detection model, and the comparison results between the labeling results of applying the object detection model and the actual labeling are shown in Figure 3 and Figure 4, respectively. It can be seen that the positions and categories annotated by the detection model are almost identical to the actual annotations, and the detection model is more delicate and accurate in target framing.

Finally, according to the position coordinates of the prediction box, the corresponding images of the face, the left stripes, and the right stripes are obtained by segmentation and cropping, as shown in Figure 5.

#### 2.2.4. Performance Comparison with Other Object Detection Models

We also conducted comparative experiments with models such as Faster R-CNN, SSD, and YOLOv5 using the same dataset and training parameters. The comparison results of precision (P), recall (R), and mAP@0.5 are shown in Table 1.

From the table, it is evident that the YOLOv5 model exhibits the best detection performance, ranking first in both precision and recall among all models, and also demonstrating the highest detection precision. Through the aforementioned comparative experiments, the effectiveness of the model constructed in this paper has been fully validated.

## 3. The Improved InceptionResNetV2 Model

### 3.1. Selection of the Base Network

Due to factors such as lighting conditions, background occlusion, and image clarity, wild Amur tiger images make individual recognition based on facial and stripe features quite challenging, to the extent that even human visual recognition can result in misjudgments, necessitating the selection of a network that has strong fine-grained texture feature expression, small computational parameter volume, and fast network convergence speed.

Inspired by the performance optimization principle of the ResNet residual network, InceptionResNet [34], introduced alongside InceptionV4 in 2016, integrates Inception modules into residual blocks. This confers the following advantages to the InceptionResNet network: it breaks down large convolutional kernels into multiple smaller ones, significantly reducing the computational volume of parameters; it maintains rich feature expression capabilities while achieving a balance between network width and depth; and it uses residual connections to deepen the network to mitigate the vanishing and exploding gradient problems, as well as to improve network optimization efficiency, thus accelerating network convergence speed. The latest InceptionResNetV2 model sets different hyperparameters and changes the number of channels based on InceptionResNetV1, and its Stem network structure is identical to that of InceptionV4. Research results indicate that InceptionResNetV2 has a similar accuracy rate to InceptionV4, but the Inception structure with residual connections allows it to have a faster convergence speed compared to InceptionResNetV1. Particularly for the individual identification of the Amur tiger, where there are many details in the face and stripes, and texture features need to be extracted at different granularities, the network structure of InceptionResNetV2 is sufficiently complex, with strong expression capabilities and rich spatial features. It can also relatively reduce the large number of parameters generated by features such as stripes, and is particularly adept at analyzing such problems. Therefore, this paper adopts InceptionResNetV2 as the foundational network.

InceptionResNetV2 introduces Inception modules into residual blocks, resulting in three new InceptionResNet modules, A, B, and C, where pooling operations are replaced by residual connections, as shown in Figure 6.

### 3.2. Improved InceptionResNetV2 Model

The InceptionResNetV2 model itself performs well in image classification tasks. However, recognizing individual Amur tigers in real-world scenarios can be challenging due to background interference, tree shadows, and other factors, resulting in high computational complexity and difficulty. To fully leverage the rich pattern distribution of Amur tigers, enhance the expression capability of pattern features at different granularities, and minimize overfitting during the model training process, this paper further improves the InceptionResNetV2 model by fully utilizing its multiple advantages. The specific methods are as follows.

#### 3.2.1. Methodology

Adding Dropout Layer

Dropout [35], proposed by Hinton in 2012, is a widely used strategy for solving model overfitting problems in deep learning. It will randomly select some neurons to be temporarily discarded as it propagates, which is equivalent to changing the model from one to a combination of several different models, making the model more generalized.

According to experience, in the deep network, the maximum regularization effect can be obtained by using a dropout rate of about 0.5. In shallow networks, the dropout rate should be lower than 0.2, because a large dropout rate will cause too much input data to be lost and have a large impact on the model. Therefore, this paper trained models with dropout rates of 0, 0.1, 0.2, 0.3, 0.4, and 0.5, and selected the optimal dropout rate through experimental verification.

2.Introduction of CBAM

In 2018, Sanghyun Woo et al. proposed a lightweight structure CBAM [36] based on attention mechanism, which is an attention module specially designed for convolutional neural networks and can widely improve the representation ability of CNN. CBAM is a hybrid attention mechanism, which processes a channel attention [37] module and a spatial attention module for the input feature layer.

CBAM is a relatively independent module that can be directly applied to most models to achieve plug-and-play effects. However, the features that the attention mechanism focuses on are not necessarily important features, and sometimes they may affect the effect of feature extraction and classification accuracy of the model. Therefore, the location of the CBAM module in the convolutional neural network is particularly important. In order to use the weight of the pre-trained model to improve its recognition accuracy with a small amount of data, and to compare it with other existing models, this paper chose to add the attention mechanism after the last Inception-ResNet-C module, so that the model can better extract features, so as to improve the model recognition accuracy.

#### 3.2.2. Definition of the Improved InceptionResNetV2 Model

Firstly, we define the input layer of the model. The fixed input size used in this model during the training phase is (299, 299), and since the samples are color images, the number of channels in the input layer is three. Assuming the batch size is m, the input layer sample set can be represented as Formula (1):(1)$$X\in R^{299\times 299\times 3\times m}.$$

Then, the shallow Stem structure is defined to quickly reduce the resolution of the feature map. The specific calculation formulas are shown in (2)~(8).
(2)$$C_{1}=ReLU(BN(Conv(X,w_{1}))),$$
(3)$$C_{2}=ReLU(BN(Conv(C_{1},w_{2}))),$$
(4)$$C_{3}=ReLU(BN(Conv(C_{2},w_{3}))),$$
(5)$$X_{1}=Maxpool(C_{3},size=3,stride=2),$$
(6)$$C_{4}=ReLU(BN(Conv(X_{1},w_{4}))),$$
(7)$$C_{5}=ReLU(BN(Conv(C_{4},w_{5}))),$$
(8)$$X_{2}=Maxpool(C_{5},size=3,stride=2),$$

Then, the extracted feature map is input into the Inception-ResNet-A module, in which $X_{2}$ is calculated on the four branches respectively, and the final results are concatenated along the channel dimension. The specific process can be represented by Equations (9)–(17):(9)$$A_{1.1}=ReLU(BN(Conv(X_{2},w_{1.1}))),$$
(10)$$A_{2.1}=ReLU(BN(Conv(X_{2},w_{2.1}))),$$
(11)$$A_{2.2}=ReLU(BN(Conv(A_{2.1},w_{2.2}))),$$
(12)$$A_{3.1}=ReLU(BN(Conv(X_{2},w_{3.1}))),$$
(13)$$A_{3.2}=ReLU(BN(Conv(A_{3.1},w_{3.2}))),$$
(14)$$A_{3.3}=ReLU(BN(Conv(A_{3.2},w_{3.3}))),$$
(15)$$A_{t}=Concat(A_{1.1},A_{2.2},A_{3.3}),$$
(16)$$A_{1}=ReLU(BN(Conv(A_{t},w_{t}))),$$
(17)$$X_{A}=ReLU(A_{1}※X_{2}).$$

After being processed five times by the Inception-ResNet-A module, the feature maps need to be transmitted to the Reduction-A module to reduce the feature map size from 35 × 35 to 17 × 17 to decrease computational complexity. The specific process can be represented by Equations (18)–(23):(18)$$R_{1.1}=ReLU(BN(Conv(X_{A},{wr}_{1.1}))),$$
(19)$$R_{2.1}=ReLU(BN(Conv(X_{A},{wr}_{2.1}))),$$
(20)$$R_{2.2}=ReLU(BN(Conv(R_{2.1},{wr}_{2.2}))),$$
(21)$$R_{2.3}=ReLU(BN(Conv(R_{2.2},{wr}_{2.3}))),$$
(22)$$R_{3}=maxpool(X_{A},size=3,stride=2),$$
(23)$$X_{RA}=Concat(R_{1.1},R_{2.3},R_{3}).$$

The feature maps obtained from the Reduction-A module are subsequently processed 10 times through the Inception-ResNet-B module, one time through the Reduction-B module, and five times through Inception-ResNet-C module. The calculation process is similar to that of module A, so the intermediate calculations are not elaborated here. The output $X_{c}$ from the last Inception-ResNet-C module is fed into the CBAM module with a dual-attention mechanism for computation. The specific calculation can be represented by Equation (24):(24)$$Y_{1}=F_{atts}(F_{attc}(X_{c}))⊙X_{c},$$
where ⊙ denotes element-wise multiplication, and $Y_{1}$ represents the output of more refined feature maps. Then, $Y_{1}$ is sequentially passed through the global average pooling layer and the dropout layer for computation, and subsequently forwarded to the final convolutional layer. The final extracted feature vector is classified using the Softmax function to obtain the ultimate classification result, which is represented by Equation (25):(25)Y=softmax(Conv(Y,w)).

After the aforementioned process, the Amur tiger individual recognition model constructed in this paper is finally obtained, and its structure is shown in Figure 7.

## 4. Experimental Results and Analysis

### 4.1. Data Preprocessing

This paper performs individual recognition based on the image segmentation results from Section 2, with three categories: face, left-side stripes, and right-side stripes. The number of images and the number of Amur tigers corresponding to each category are shown in Table 2.

Since the fixed input size of the InceptionResNetV2 model is 299 × 299 pixels, and most images in the dataset used in this paper have different aspect ratios, it is necessary to resize the images to squares and scale them. To avoid distortion and retain more information, this paper adopts the method of padding according to the long side, as shown in Figure 8.

To avoid overfitting due to the small dataset size, this paper also introduces data augmentation techniques to expand the dataset, including adjusting brightness, contrast, hue, and saturation, and applying blur to images. The augmentation effects are shown in Figure 9.

After augmenting the dataset, it is randomly divided into training and test sets in a 4:1 ratio.

### 4.2. Pre-Trained Model Loading

Since the sample size of the Amur tiger dataset used in this paper is relatively small, to improve the classification accuracy of the model and effectively avoid overfitting, this paper further introduces transfer learning technology, using the InceptionResNetV2 model parameters trained on the ImageNet dataset. Given the distinct features of the Amur tiger dataset compared to ImageNet, we load the InceptionResNetV2 pre-trained model from tensorflow.keras.applications and include its full architecture in our training to leverage transfer learning effectively.

### 4.3. Model Training

#### 4.3.1. Experimental Environment Setup and Training Parameters

The software and hardware environment configurations employed in the experiments of this paper are shown in Table 3.

The training parameters are set as follows:(1)Number of iterations: epoch = 20;(2)Batch size: batch_size = 16;(3)Initial learning rate: lr0 = 0.001;(4)optimizer: optimizer = Adam;(5)loss function: loss = Cross-entropy.

#### 4.3.2. Model Training Process

The training process of the improved InceptionResNetV2 model is shown in Figure 10. By applying the preprocessed Amur tiger images as the original input to the model, a series of convolutional and pooling layers are stacked in the Stem structure to obtain shallow feature maps of the images. The InceptionResNet module extracts different-sized feature maps through parallel convolution and merges them to further extract deep features of the images. The CBAM module focuses on important features to obtain more refined feature maps. Then, the extracted features are input into the global average pooling layer to calculate the average value of all pixel values in each channel. Subsequently, a certain proportion of neurons are randomly dropped, and the Softmax classification layer outputs the classification results. After completing the forward propagation process, the loss is calculated and the parameters are updated according to the backpropagation algorithm. These steps are repeated until the loss is below the pre-set threshold and no longer changes, indicating the end of training.

### 4.4. Comparison of Results

#### 4.4.1. Comparison and Analysis of Different Dropout Rates

In this paper, the dropout layer is added after the global average pooling layer of the original InceptionResNetV2 model. To illustrate the impact of different dropout rates on the training results and select the optimal dropout rate, we trained the model with dropout rates of 0, 0.1, 0.2, 0.3, 0.4, and 0.5, respectively. The experimental results show that the optimal dropout rate for the facial, left stripe, and right stripe data is consistent. Taking the facial data as an example, the training results are shown in Figure 11.

Each epoch of training has a Top1 accuracy, and we generally compare the maximum Top1 accuracy, which is the maximum value of Top1 accuracy over 20 rounds of training. However, due to possible deviations and small differences in the maximum Top1 accuracy, this paper specifically defines the fifth Top1 accuracy for collaborative comparison, which is the fifth-highest Top1 accuracy in 20 rounds of training. The maximum Top1 accuracy and the fifth-highest Top1 accuracy of several methods are combined, and the results are shown in Table 4 below.

From the table analysis, we can see that when the dropout rate is 0.4, the fifth-highest Top1 has the highest precision, so this paper uses a dropout rate of 0.4.

#### 4.4.2. Comparison and Analysis of Training Effectiveness with Different Attention Mechanisms

We added an attention mechanism after the last Inception-ResNet-C module in the original InceptionResNetV2 model. To verify the impact of different attention mechanisms on model training effectiveness, we experimented with the SE channel attention mechanism, the ECA [38] module, and the CBAM dual attention mechanism separately. Similarly, the experimental results for the facial, left stripe, and right stripe regions are consistent. Taking the face data as an example, the training results are shown in Figure 12.

The specific results of accuracy are shown in Table 5. As indicated in the table, when the InceptionResNetV2 model is augmented with the CBAM module, it achieves the highest recognition accuracy. Moreover, it exhibits the fastest convergence speed among all methods incorporating attention mechanisms, along with higher stability.

#### 4.4.3. Comparison with Other Models

After adding the dropout layer and CBAM module to the original InceptionResNetV2 model, the network structure is shown in Table 6 below.

To evaluate the performance of the Amur tiger individual recognition method proposed in this paper, we trained several different network models, including VGG19, ResNet50, ResNet152, InceptionV3, InceptionResNetV2, and the improved InceptionResNetV2 with the same dataset and training parameters, and tested all models on the same testing dataset. Taking the facial data as an example, the accuracy and loss comparison results of each model on the testing dataset are illustrated in Figure 13 and Figure 14, respectively.

From Figure 13, it can be seen that, in terms of accuracy variation, the performance of the proposed improved model consistently outperforms the first four models from the initial stage to the final stage. Although sometimes it is on par with the InceptionResNetV2 model, the accuracy changes of the improved model are relatively smoother.

From Figure 14, it can be observed that, in terms of the change in loss values, the model proposed in this paper converges faster and exhibits higher stability. As indicated by the specific accuracy values in Table 7, the improved InceptionResNetV2 model achieves the highest recognition accuracy, with an improvement of 1.38% compared to the original InceptionResNetV2 model. Through the aforementioned comparative experiments, the effectiveness of the proposed model in this paper has been fully validated.

#### 4.4.4. Comparison and Analysis of Different Physiological Characteristic Parts

The improved model is applied to different physiological features of Amur tigers, and the test accuracy data is compared with the original model, as shown in Table 8 below.

As shown in the table, the recognition accuracy of the left-side stripes is the highest, and the accuracy for the right-side stripes is slightly higher than that of the face. Since most Amur tigers only show their side faces, the recognition accuracy for the face should be lower than that of the left-side and right-side stripes. However, almost every image contains a facial part, and the facial dataset is somewhat larger than the right-side stripe dataset. Therefore, due to the interaction of these two factors, the difference in accuracy between the two is not significant.

It is important to note that in the actual process of individual recognition conducted in this study, if a picture contains multiple physiological feature parts of an Amur tiger, the part with the highest recognition accuracy is taken as the final identification result. For example, if a picture shows both the face and the right stripe of an Amur tiger, based on experimental results, the order of recognition accuracy from high to low for different parts is left stripe > right stripe > face. Therefore, the result of recognizing the right stripe will be considered as the final identification conclusion.

### 4.5. Discussion

The individual identification of Amur tigers presents a unique challenge due to the intricate details of facial features and stripes, which require feature extraction at varying granular levels. The InceptionResNetV2 network is well-suited for this task due to its complex network structure that offers robust representation capabilities and rich spatial features. Its design allows for effective analysis of stripe patterns while reducing the number of parameters typically associated with such detailed features, making it particularly adept at addressing these issues.

In addition to the Amur tiger, many other animals such as golden snub-nosed monkeys, Amur leopards, giant pandas, zebras, and lions also possess unique body or facial stripes. When identifying the species or individuals of these animals, fine-grained feature recognition is similarly required. Therefore, the method proposed in this paper can be applied to similar recognition scenarios, making it suitable for identifying a wider range of animal species or individuals. This demonstrates that our method has broad prospects for promotion and application.

To adapt and refine our method for broader use in individual identification across various species, we can employ similar strategies tailored to different animal characteristics. For example, to apply our approach to the indentification of zebras, similarly, we could start by using the YOLOv5 model to detect and segment specific stripe patterns on different body parts, and then carry out individual identification through the InceptionResNetV2 network. Finally, the corresponding model could be applied to a specialized zebra dataset for training and validation, ultimately resulting in a high-precision model capable of accurately identifying different body part features of individual zebras. 

Additionally, the improved method proposed in this paper can be applied to other network models such as EfficientNet to obtain models with higher recognition accuracy. Although the average recognition accuracy of the proposed method in this paper reaches 95.36%, there are still a few images that were not accurately recognized. Based on the characteristics of the dataset and the recognition process, the reasons for these misidentifications include the high similarity of certain body parts, interference from the angle and lighting conditions in which the photos were taken, and the relatively limited number of Amur tigers in the dataset, which leaves room for further improvement in the model’s accuracy. Therefore, in future research, we could consider incorporating techniques such as multi-scale feature fusion and multi-head attention mechanisms to enhance the recognition accuracy and efficiency of the model, providing better solutions for individual identification across various species. 

Despite the significant advantages achieved by the proposed method in terms of recognition accuracy, it also has some limitations. For instance, the dataset used for training and validating the recognition model is still quite limited. To further improve the model’s accuracy and generalization, it is necessary to collect more diverse and comprehensive datasets for training and use more image augmentation techniques to expand the dataset, including geometric distortions. Additionally, the method was only tested on the Amur tiger dataset. While the results are promising, its applicability and generalizability to other animal species still need further verification. Enhancing the model’s generalization capabilities and ensuring its robustness across different contexts are critical areas for future research.

## 5. Conclusions

As the Amur tiger is a flagship species of precious and endangered wildlife, the accurate identification of individual Amur tigers holds significant symbolic importance for biodiversity conservation. This paper focuses on the methods of object detection and individual recognition of Amur tigers based on convolutional neural networks. Initially, the YOLOv5 model is employed for object detection on the Amur tiger dataset, obtaining the facial, left-side stripe, and right-side stripe data of individual tigers, followed by image segmentation. To further improve the individual recognition performance of Amur tigers, this paper proposes an improved model based on InceptionResNetV2, which incorporates dropout regularization to prevent overfitting and introduces a dual-attention mechanism to strengthen the feature representation capabilities at different levels. Then, transfer learning is performed by employing a pre-trained model on the ImageNet dataset to improve model training efficiency. Finally, comparative experiments are conducted on the open large-scale wildlife dataset ATRW, comparing the improved InceptionResNetV2 model with other classic image recognition models. The experimental results demonstrate the superior performance of the proposed model, validating its effectiveness and practicality.

Our work provides a meaningful exploration for the accurate identification of individual wild animals. In our future work, we aim to further enhance aspects such as dataset richness, model performance, and system functionality.

## Figures and Tables

**Figure 1 animals-14-02312-f001:**
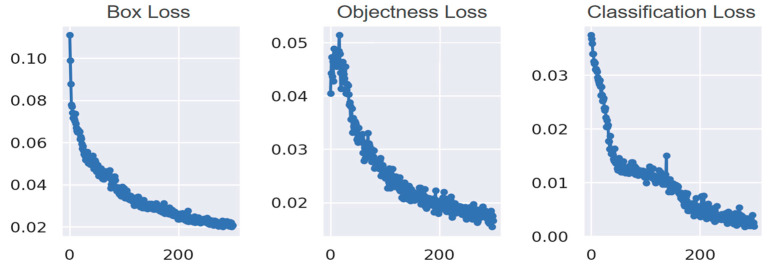
Rectangle box loss, objectness loss, and classification loss in the training proces.

**Figure 2 animals-14-02312-f002:**
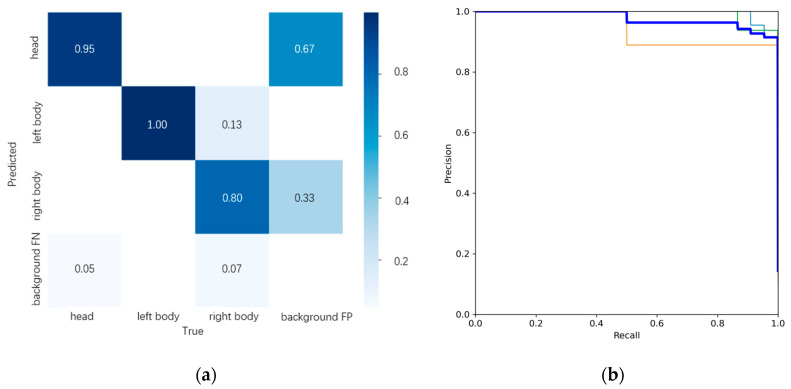
Validation indexes. (**a**) confusion matrix diagram; (**b**) the P-R curve. The P-R curve colors represent the following categories: Light Blue line - head, Orange line - left body, Green line - right body, Deep Blue line - all classes.

**Figure 3 animals-14-02312-f003:**
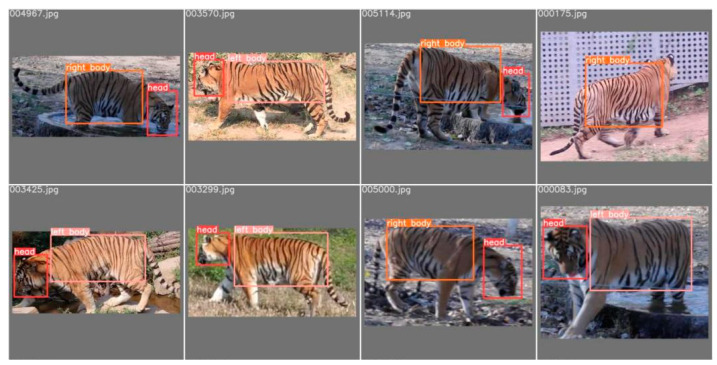
The actual annotations of the validation dataset.

**Figure 4 animals-14-02312-f004:**
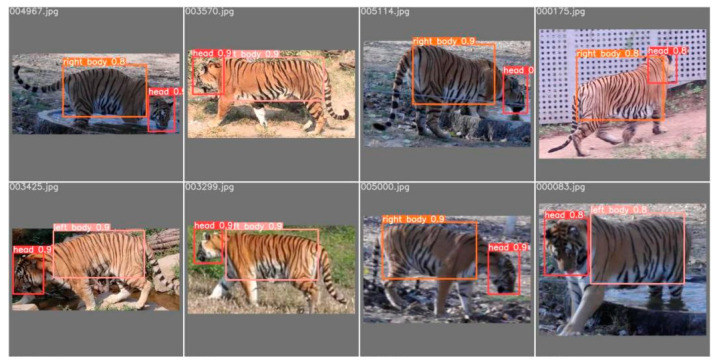
The annotation results of the validation dataset by the object detection model.

**Figure 5 animals-14-02312-f005:**
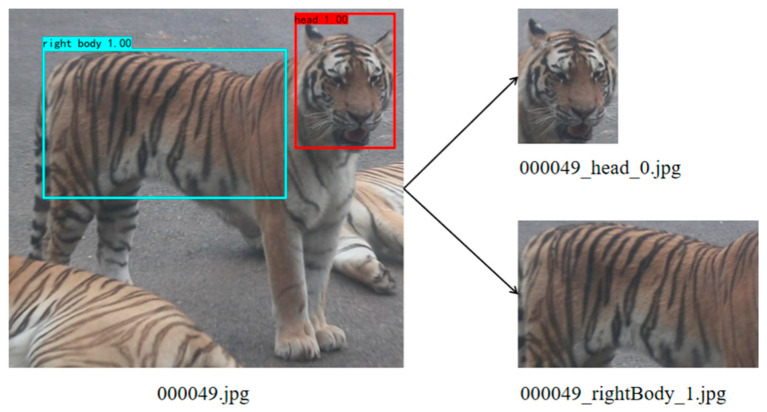
An example of segmentation results.

**Figure 6 animals-14-02312-f006:**
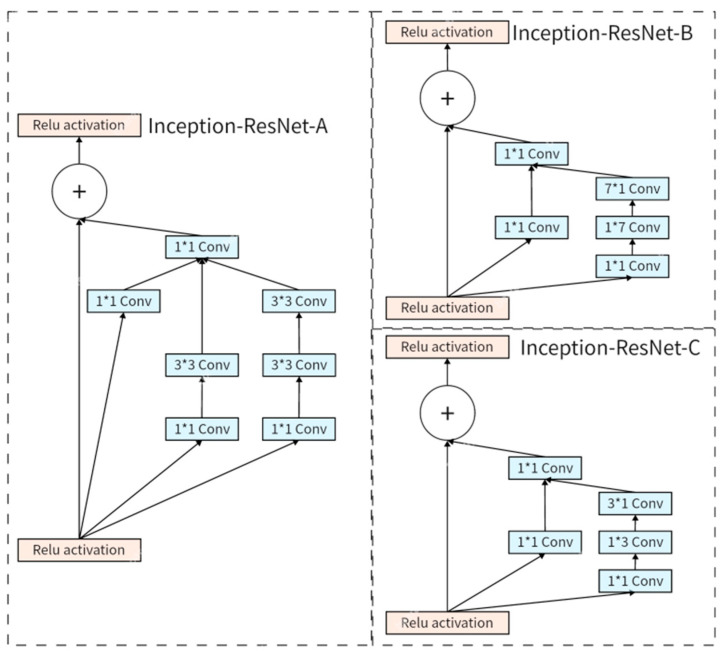
The structure of the Inception-ResNetV2 module.

**Figure 7 animals-14-02312-f007:**
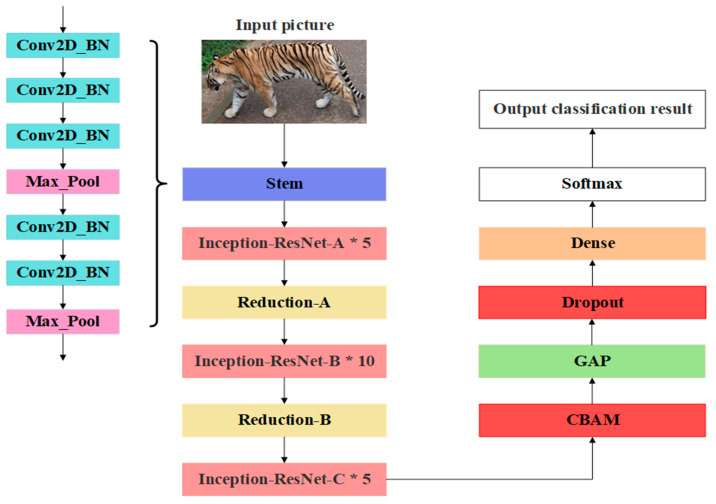
The structure of the improved InceptionResNetV2 model.

**Figure 8 animals-14-02312-f008:**
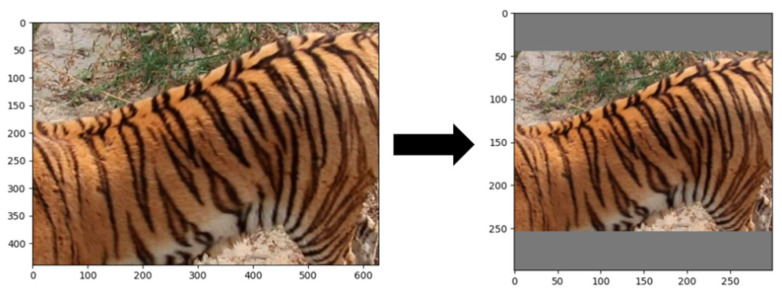
Image padding based on the long side.

**Figure 9 animals-14-02312-f009:**
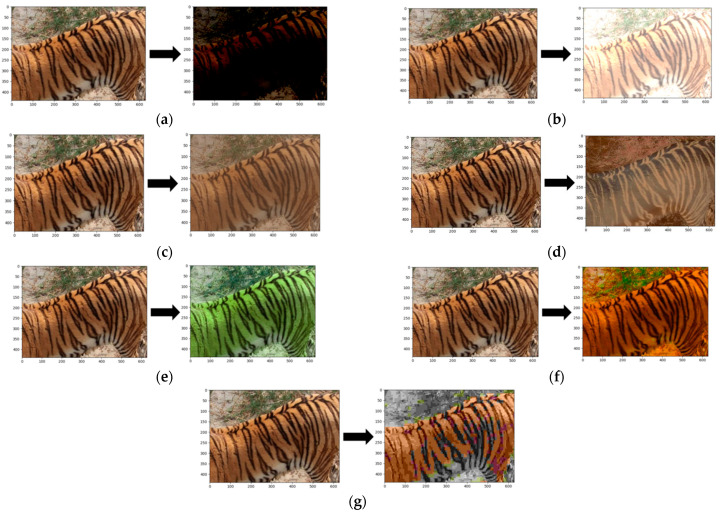
Data enhancement. (**a**) reduce brightness; (**b**) increase brightness; (**c**) reduce contrast; (**d**) increase contrast; (**e**) hue adjustment; (**f**) saturation adjustment; (**g**) blur image.

**Figure 10 animals-14-02312-f010:**
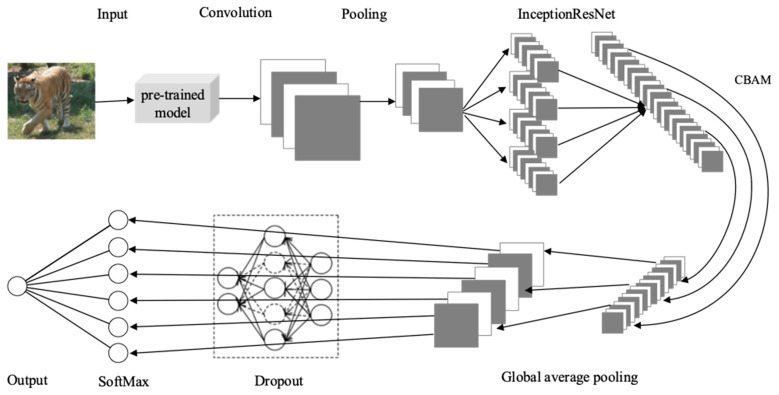
Model training process.

**Figure 11 animals-14-02312-f011:**
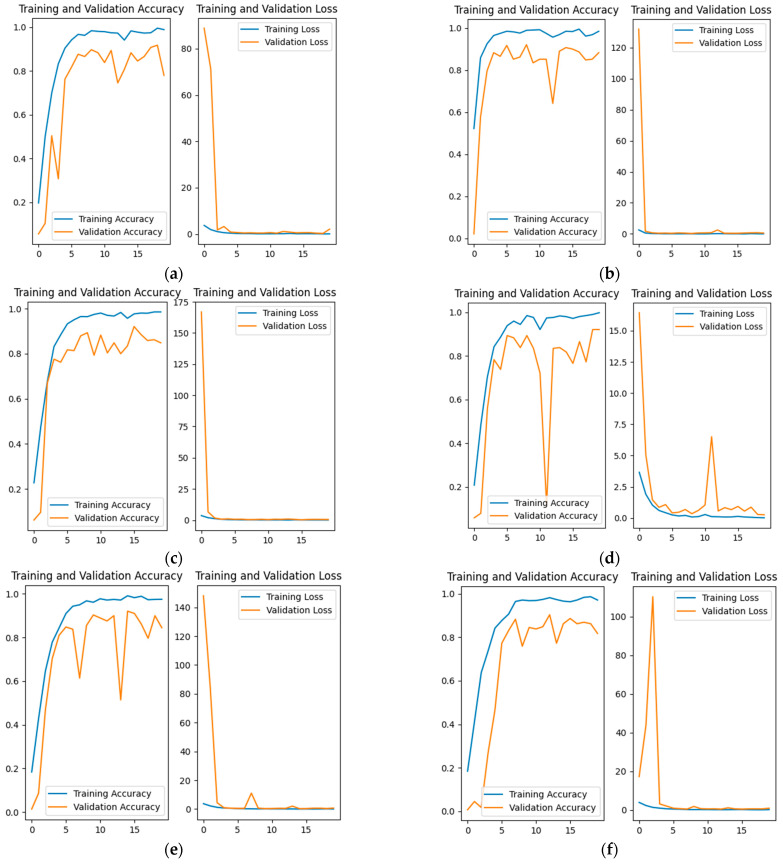
Comparison of training effectiveness with different dropout rates. (**a**) dropout rate = 0; (**b**) dropout rate = 0.1; (**c**) dropout rate = 0.2; (**d**) dropout rate = 0.3; (**e**) dropout rate = 0.4; (**f**) dropout rate = 0.5.

**Figure 12 animals-14-02312-f012:**
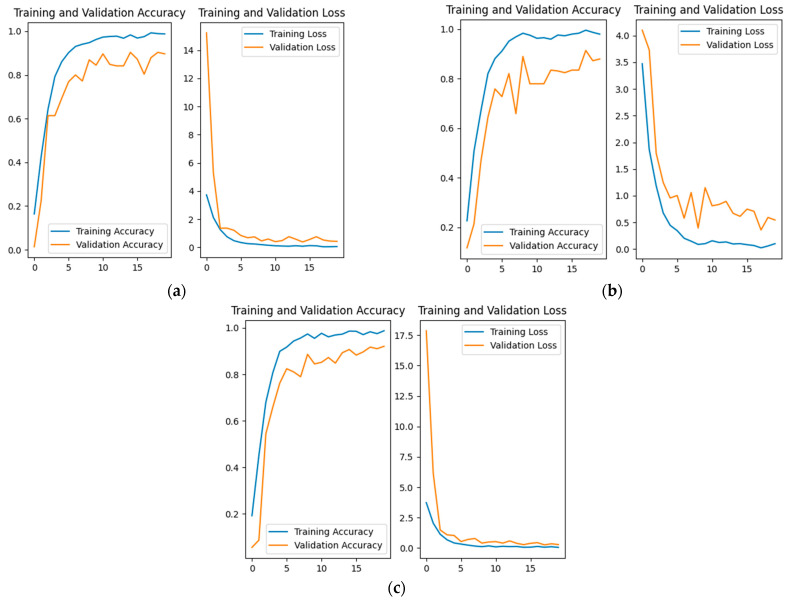
Comparison of training effectiveness with different attention mechanisms. (**a**) Adding an SE module; (**b**) adding an ECA module; (**c**) Adding an CBAM module.

**Figure 13 animals-14-02312-f013:**
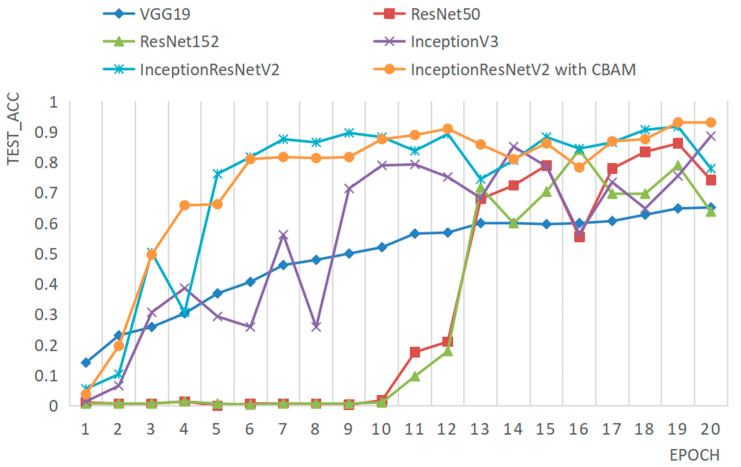
Comparison of the accuracy of different models on the same test set.

**Figure 14 animals-14-02312-f014:**
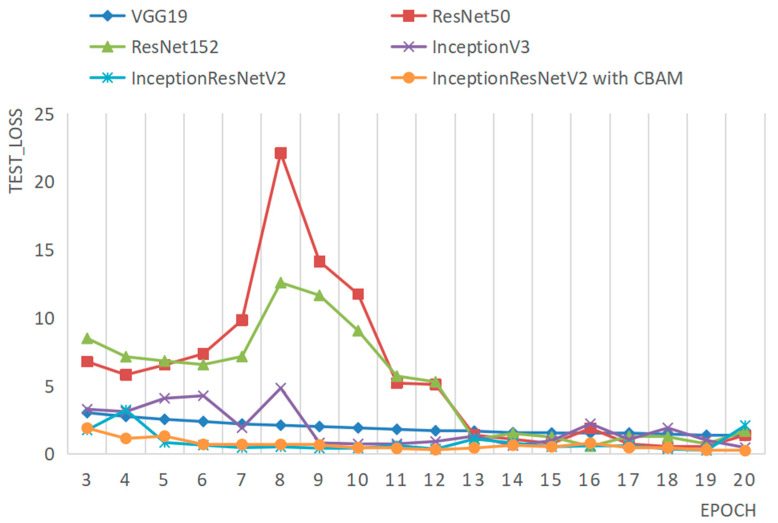
Comparison of the loss of different models on the same test set.

**Table 1 animals-14-02312-t001:** Comparison results with other models.

Model	P	R	mAP@0.5
Faster R-CNN	56.23%	94.87%	82.73%
SSD	83.29%	90.17%	90.49%
**YOLOv5**	**88.90%**	**95.00%**	**97.30%**

**Table 2 animals-14-02312-t002:** Data situation for individual recognition.

	Face	Left Stripe	Right Stripe	Complete Image
Number of pictures	1668	891	992	1887
Number of Amur tigers	107	59	66	107

**Table 3 animals-14-02312-t003:** Experimental environmental configuration and versions.

Hardware and Software Configuration	Versions
GPU	RTX 2080 Ti (11 GB)
CPU	Intel(R) Xeon(R) Platinum 8255C CPU @ 2.50 GHz
OS	Ubuntu
CUDA	11.2
TensorFlow	2.5.0

**Table 4 animals-14-02312-t004:** Experimental results with different dropout rates.

Dropout Rate	Maximum Top1 Accuracy	Fifth Top1 Accuracy
0	0.9172	0.8828
0.1	0.9207	0.8897
0.2	0.9207	0.8793
0.3	0.9207	0.8828
**0.4**	**0.9207**	**0.9000**
0.5	0.9034	0.8690

**Table 5 animals-14-02312-t005:** Experimental results of different attention mechanisms.

Attention Mechanism	Accuracy Rate	Training Time/s
Original model	0.9172	460
SE	0.9034	474
ECA	0.9138	472
**CBAM**	**0.9207**	**469**

**Table 6 animals-14-02312-t006:** The improved network structure of the InceptionResNetV2 model.

Layer (Type)	Output Shape	Param
inception_resnet_v2 (Functional)	(None, 8, 8, 1536)	54,336,736
cbam_block (cbam_block)	(None, 8, 8, 1536)	589,922
global_average_pooling2d_1 (GlobalAveragePooling2D)	(None, 1536)	0
dropout (Dropout)	(None, 1536)	0
dense_2 (Dense)	(None, 107)	164,459

**Table 7 animals-14-02312-t007:** Comparison of accuracy and training time of different models.

Model	Accuracy Rate	Training Time/s
VGG19	0.6517	104
ResNet50	0.8621	149
ResNet152	0.8414	340
InceptionV3	0.8862	194
InceptionResNetV2	0.9172	460
**InceptionResNetV2 with CBAM**	**0.9310**	**469**

**Table 8 animals-14-02312-t008:** Experimental results of different physiological characteristics.

Model	Head	Left Body	Right Body
InceptionResNetV2	0.9172	0.9937	0.9186
InceptionResNetV2 with CBAM	0.9310	0.9937	0.9360

## Data Availability

The ATRW dataset is available at https://lila.science/datasets/atrw (accessed on 24 February 2024).

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
