# Peer review of "Amur Tiger Individual Identification Based on the Improved InceptionResNetV2"

_animals, 2024, doi:10.3390/ani14162312_

Round 1

Reviewer 1 Report

Comments and Suggestions for Authors

The authors present an interesting study on individual detection of Amur Tiger using InceptionResNetV2 Model. The bright side of the manuscript is that to provide important results detection and monitoring of Amur Tiger and it provides improved model for individual detection. In this context, the manuscript like to contribute different fields such as animal ecology and animal behaviour. However, some parts of the manuscript need to be improved and restructured. Therefore, I would like to make some suggestions to improve the quality of the paper as below:

Line 12: “flagship wildlife individuals” -> “individuals of flagship wildlife species”.

Line 13: Please add the scientific name of the species.

Lines 32-38: “As an important part of the ecosystem, wildlife is closely related to the maintenance of ecological balance and stability. However, in recent years, with the continuous expansion of human production and social activities, the habitats and living spaces of animals have been continuously encroached upon and destroyed, leading to many animal species being on the brink of extinction or in a critically endangered state, posing a severe threat to the biodiversity of China and even the world. Decisive and feasible conservation measures must be taken.” Please rephrase here. Also, references are needed.

Lines 41-42: “Comparatively, most studies on wildlife behavior and ecology focus on individual identification rather than species recognition”. I do not agree with the authors. Individual recognition is important to understand in many aspects such as density, space use, predator-prey relationships, understanding species behaviours etc. However, species recognition is in core of many ecological studies. In this context, such sentence or similar sentence can be replaced by this sentence “Recently studies showed that individual identification becoming more important since it helps us to understand density, space use, behaviours etc. of different species (references)”. Please add several references such as doi: 10.1016/j.ecoinf.2022.101919, 10.3390/ani10071207, 10.1016/j.procs.2023.12.065, 10.3390/ani11061723.

Line 115: “2. Amur tiger stripe part detection with YOLOV5” This heading should be under the Methods Section. Also, the manuscript should have Introduction, Methods, Results, Discussion and Conclusion sections. In my opinion, the authors reorganize/restructure the manuscript by adding the Methods, Results, Discussion sections.

Discussion section can be enriched with a more theoretical interpretation and relate the present results with additional concepts. For instance, the study results can be discussed with similar studies from different methods and animal species in the broader context. Moreover, the limitations of the study should be explained in discussion or conclusion section.

Comments on the Quality of English Language

Some sentences (given in the comments) should be rephrased. 

Reviewer 2 Report

Comments and Suggestions for Authors

Animals review 7/2024

Research on Individual Recognition Method of Amur Tiger Based on Improved InceptionResNetV2 Model

A brief summary (one short paragraph) outlining the aim of the paper, its main contributions and strengths.

Considering their classification as an endangered flagship species, the ability to recognize individual Amur tigers is essential. Performance of the YOLOv5 model for feature detection, framing, and labeling is impressive for the Amur tiger dataset. Description of the base network structure and methodology using dropout and CBAM is well articulated, as are the resulting recognition rates.

General concept comments

            The discussion on image augmentation addresses only point based modifications such

as contrast and brightness. While those augmentations make sense, additional

augmentations for common geometric distortions (particularly those mimicking gradual changes in camera angle) seem appropriate.

Some discussion of the characteristics of images which are not accurately classified, would strengthen the analysis.

Specific comments referring to line numbers, tables or figures that point out inaccuracies within the text or sentences that are unclear. These comments should also focus on the scientific content and not on spelling, formatting or English language problems, as these can be addressed at a later stage by our internal staff.

      Title of paper could be shortened/improved

      The reference to “left stripes” and “right stripes” is somewhat ambiguous before Figure 3 and 4 are encountered. Stripes on left side of animal or left side stripes might clarify initially. (17,23,135)

      (51) “specific fur stripes” seems more appropriate terminology for distinguishing between species. Distinct perhaps?

      (88) pixel quality - does this refer to image resolution?

      Figure 1. The labels above the graphs do not correspond with the figure captions - the same terminology should be used. Also, markers on the line graph might be omitted to improve clarity of the graph values.

      (183) “fringe” is an unexpected term here.

      Figure 9. Three images in the figure are incorrect. The result shown for “reduce contrast” looks like a digital negative.The result shown for increased contrast looks like a “reduce contrast.” Blur image looks like downsampling. Not sure why tone adjustment is used for augmentation

      (319-323) Something redundant here

      Table 3. The term parameter is misleading here. Perhaps model/version?

      Table 4. Header should be Dropout rate not discard rate, for consistency

      Figures 13 and 14 are well presented and described.

Comments on the Quality of English Language

Paper is clear and understandable but has a number of minor syntax errors. Many instances of missing article (the/a) and lack of plural form (s). Occasionally a word choice is made that is not quite the best word for the point being made. A small number of typos.

Round 2

Reviewer 1 Report

Comments and Suggestions for Authors

The authors improved the manuscript with the previous comments. I suggest only following improvements;

Line 25: Panthera tigris altaica (hereafter referred to as the Amur tiger -> Amur tiger (Panthera tigris altaica). Scientific names should be italic, and the common name should written at first and second scientific name.

Line 455: Discussions -> Discussion